# Effect of Fluoroquinolone Use in Primary Care on the Development and Gradual Decay of *Escherichia coli* Resistance to Fluoroquinolones: A Matched Case-Control Study

**DOI:** 10.3390/antibiotics11060822

**Published:** 2022-06-18

**Authors:** Peter Konstantin Kurotschka, Chiara Fulgenzio, Roberto Da Cas, Giuseppe Traversa, Gianluigi Ferrante, Orietta Massidda, Ildikó Gágyor, Richard Aschbacher, Verena Moser, Elisabetta Pagani, Stefania Spila Alegiani, Marco Massari

**Affiliations:** 1Department of General Practice, University Hospital Wuerzburg, Josef-Schneider-Str. 2, 97080 Wuerzburg, Germany; gagyor_i@ukw.de; 2Pharmacy Unit, IRCCS Regina Elena National Cancer Institute and San Gallicano Institute, 00128 Rome, Italy; fulgenzio.chiara@gmail.com; 3Pharmacoepidemiology and Pharmacovigilance Unit, National Centre for Drug Research and Evaluation, Italian National Institute of Health (ISS), 00161 Rome, Italy; roberto.dacas@iss.it (R.D.C.); g.traversa@aifa.gov.it (G.T.); stefania.spila@iss.it (S.S.A.); marco.massari@iss.it (M.M.); 4Italian Medicine Agency (AIFA), 00187 Rome, Italy; 5Azienda Ospedaliera-Universitaria Città della Salute e della Scienza di Torino, 10126 Turin, Italy; gianluigi.ferrante@cpo.it; 6Department of Cellular, Computational and Integrative Biology, Center of Medical Sciences (CISMed), University of Trento, 38122 Trento, Italy; orietta.massidda@unitn.it; 7Health Service of the Autonomous Province of Bolzano/Bozen, 39100 Bolzano/Bozen, Italy; richard.aschbacher@sabes.it (R.A.); verena.moser@provincia.bz.it (V.M.); elisabetta.pagani@sabes.it (E.P.)

**Keywords:** drug resistance, bacterial, antimicrobial resistance, anti-bacterial agents, primary care, *Escherichia coli*, quinolones, fluoroquinolones, information storage and retrieval

## Abstract

The reversibility of bacterial resistance to antibiotics is poorly understood. Therefore, the aim of this study was to determine, over a period of five years, the effect of fluoroquinolone (FQ) use in primary care on the development and gradual decay of *Escherichia coli* resistance to FQ. In this matched case–control study, we linked three sources of secondary data of the Health Service of the Autonomous Province of Bolzano, Italy. Cases were all those with an FQ-resistant *E. coli* (QREC)-positive culture from any site during a 2016 hospital stay. Data were analyzed using conditional logistic regression. A total of 409 cases were matched to 993 controls (FQ-sensitive *E. coli*) by the date of the first isolate. Patients taking one or more courses of FQ were at higher risk of QREC colonization/infection. The risk was highest during the first year after FQ was taken (OR 2.67, 95%CI 1.92–3.70, *p* < 0.0001), decreased during the second year (OR 1.54, 95%CI 1.09–2.17, *p* = 0.015) and became undetectable afterwards (OR 1.09, 95%CI 0.80–1.48, *p* = 0.997). In the first year, the risk of resistance was highest after greater cumulative exposure to FQs. Moreover, older age, male sex, longer hospital stays, chronic obstructive pulmonary disease (COPD) and diabetes mellitus were independent risk factors for QREC colonization/infection. A single FQ course significantly increases the risk of QREC colonization/infection for no less than two years. This risk is higher in cases of multiple courses, longer hospital stays, COPD and diabetes; in males; and in older patients. These findings may inform public campaigns and courses directed to prescribers to promote rational antibiotic use.

## 1. Introduction

Antimicrobial resistance (AMR), mainly promoted by antibiotic use, is an urgent public health issue [1,2,3]. Its notable health and economic burdens are of global concern and have resulted in international research and policy efforts with the aim of reducing antimicrobial consumption [4,5]. One area of interest is antibiotic use in primary care, since drug resistance of pathogens causing community-acquired infections is widespread and primary care physicians are responsible for the vast majority of antibiotic prescriptions issued in the human health sector, mainly for respiratory and urinary tract infections [6,7]. Previous studies have shown a strong association between antibiotic use in primary care and the emergence of AMR, with a stronger association observed when antibiotic use is recent [8,9,10]. What is still poorly understood is to what extend a reversal of existing AMR may occur due to a reduction in antimicrobial use and how much time the return of antibiotic susceptibility takes in different bacterial species and after the use of specific antibiotics [8]. In a previous study conducted in the Autonomous Province of Bolzano, located in northern Italy, we found that, over a five-year period, the risk of developing a community-acquired infection due to a third-generation cephalosporin-resistant (3GC) *Escherichia coli* increases significantly in patients who were previously exposed to antibiotics. The highest risk was observed when antibiotics were taken in the last 12 months and for greater cumulative exposures to any antibiotic, as well as to 3GC [11]. Improving our knowledge of the resistance decay of different bacterial species and after exposure to different antibiotics is critical to inform antibiotic stewardship interventions and clinical decisions with the aim of minimizing AMR worldwide [12,13]. To the best of our knowledge, no study has examined the long-term effect of fluoroquinolone (FQ) use on the development and decay of FQ resistance in *E. coli* in individual patients, despite its high prevalence and the growing resistance rates, which are causing a progressive loss of efficacy of FQs in many clinical settings [14]. Therefore, we set up the present study to determine the influence of outpatient FQ use on the development and decay of FQ resistance in *E. coli*.

## 2. Results

Out of the 1342 patients included in the analyses, specimens derived from urine cultures accounted for 73%, followed by blood cultures, which accounted for 9%.

Within the five years of the study, 409 cases and 933 controls, with a ratio of 2.3 controls per case, were included. The sample characteristics and the results of the univariate regression analysis are shown in Table 1. 

The univariate analysis showed that receiving at least one FQ prescription in the year preceding the diagnosis of resistance was associated with a higher risk of being colonized or infected with QREC (OR 3.87, 95% CI 2.88–5.18, *p* < 0.0001) than receiving an antibiotic prescription in the preceding 2nd year (OR 2.72, 95% CI 2.04–3.64, *p* < 0.0001) or in the preceding 3rd, 4th or 5th year (OR 1.90, 95% CI 1.49–2.44, *p* < 0.0001). Consistently, after adjustment for relevant confounding factors (age, gender, days of hospitalization and comorbidities (COPD and diabetes)), the risk of QREC colonization/infection was highest in the first year preceding the diagnosis of resistance (OR 2.67, 95% CI 1.92–3.70, *p* < 0.0001); then, it decreased progressively (OR 1.54, 95%CI 1.09–2.17, *p* = 0.015) to become undetectable after two years (OR 1.09, 95% CI 0.80–1.48, *p* = 0.997) (Table 2).

In addition, the analysis showed that older age, male sex, longer hospital stays and being affected by diabetes mellitus and/or COPD are independent risk factors for FQ resistance in *E. coli*.

The multivariable analysis focused on FQ use in the 12 months preceding the diagnosis of resistance (Table 3) showed, consistently with the univariate analysis, that the use of FQs is strongly associated with FQ resistance in *E. coli*. After adjustment for relevant confounding factors and including exposure to any antibiotic other than FQs in the model, a clear dose–response effect could be observed: the use of FQs increases the risk of FQ resistance in *E. coli* more than fourfold if the patient was exposed to three or more courses of FQ in the previous 12 months (OR 4.21, 95% CI 2.38–7.50, *p* < 0.0001), and it decreases with lower cumulative exposures (OR 2.76, 95% CI 1.63–4.66, *p* < 0.0001 and 2.40, 95% CI 1.62–3.56, *p* < 0.0001 after two courses and one course of FQ in the previous 12 months, respectively). 

The two sensitivity analyses that excluded patients with very recent antibiotic exposures (<15 days from the ID) and patients at risk for hospital-acquired infections showed results that were consistent with those of the main analyses (Appendix A). 

## 3. Discussion

### 3.1. Summary of the Principal Findings 

This matched case–control study is the first of its kind to use administrative and routinely collected clinical data—which characterize, in a large sample of patients, the impact of previous FQ use in primary care and QREC colonization/infection over a five-year period—in a multi-database approach. In general, the risk of developing a community-acquired QREC colonization/infection increases in all those patients who had received FQs. The risk of resistance is highest in the first year after the antibiotic is taken; afterwards, it decreases progressively, becoming undetectable after 24 months. In addition, the risk of resistance is higher after greater cumulative exposures. Apart from antibiotic use, older age, male sex, longer hospital stays, COPD and diabetes mellitus are independent risk factors for FQ resistance in *E. coli*. 

### 3.2. Findings of the Present Study in Light of Previous Observations

Our findings are in accordance with those of previous observational studies. Costelloe [9] and Bhakit [8] reported a consistent amount of existing evidence of an association between the decrease in antibiotic use and the subsequent decrease in resistance. At the same time, the authors of these systematic reviews highlighted a lack of evidence regarding time intervals of more than 12 months between antibiotic use and the diagnosis of resistance. What our study adds is the finding that FQ resistance can last for up to 24 months after the selective pressure of antimicrobials is removed, a longer time than previously reported. With respect to this, Hammond et al. recently demonstrated an association between antibiotic dispensing reduction in primary care and a decrease in ciprofloxacin, trimethoprim and amoxicillin resistance in *E. coli* within the subsequent quarter [15]. In contrast to the approach used in the latter and in other studies, e.g., as in Cuevas et al. [16], we were able to quantify the impact on resistance of previous antibiotic use at an individual level, avoiding the risk of the so-called ecological fallacy. Some studies, also mostly ecological, showed persistent bacterial resistance despite reduced antibiotic use in *E. coli* and in other species [17,18], while other studies showed the opposite [10,19,20]. A possible explanation of these divergent findings is likely the fact that bacterial resistance is a complex phenomenon, with primary care prescribing being only one of its drivers [21]. Vellinga et al. compared the resistance of *E. coli* to trimethoprim and ciprofloxacin with prescriptions of these antimicrobials at the GP practice level. The authors reported that the risk of a QREC colonization/infection increased by 8% for every additional prescription of ciprofloxacin per 1000 patients [22]. Similar to what they found at a community level, we show a clear dose–response effect in the association between FQ use and FQ resistance. 

A recent systematic review by Zhu et al. identified, among others, relevant risk factors for QREC colonization/infection as previous antibiotic use (OR 2.74, 95% CI 1.92–3.92), quinolones (OR 7.67, 95% CI 4.79–12.26), diabetes mellitus (OR 1.62, 95% CI 1.43–1.83), previous hospitalization (OR 2.06, 95% CI 1.62–2.60), male sex (OR 1.41, 95% CI 1.21–1.64) and organ transplantation (OR 2.37, 95% CI 1.17–4.79) [23]. Our results are in line with these findings with regard to the above-mentioned risk factors, except for organ transplantation (data not shown), probably due to the very low prevalence of this condition in our dataset. Zhu et al. reported that older age was not found to be a risk factor of QREC colonization/infection, differently than in the present study, although the authors hypothesized that heterogeneity among studies affected this finding. Furthermore, the study identified as significant risk factors for QREC colonization/infection the presence of a urinary catheter, urinary tract abnormalities and having had previous UTIs. We were not able to verify these associations because data on these conditions were not available to us. Nonetheless, in our case, the majority (i.e., 73%) of specimens that contributed to our study were urine samples. As we could demonstrate the reversibility of FQ resistance in *E. coli*, and taking into consideration that this species is the most common urinary pathogen [24], considered to be resistant to FQ in up to 40% of cases [14], antibiotic stewardship interventions seem well placed when directed to reduce antibiotic consumption for UTI, especially in primary care settings, as suggested elsewhere [25,26,27,28]. 

### 3.3. Strengths and Limitations of the Study

A strength of the present study is the comprehensiveness of the data source, namely, the database of the regional laboratories, the database of outpatient drug prescription records and the hospital discharge record databases. 

We used data from the regional laboratories of the Autonomous Province of Bolzano to select cases that were matched to all available controls derived from the same data source. This assured comparability of cases and controls, as both were selected from the same source population, in which we did not assume the existence of any selection factor related to the exposure or outcome. Furthermore, comparability over time (namely, when the resistance was, or was not, diagnosed) was assured as we used the index date, namely, the date of the bacterial culture, as the matching variable. 

The linkage of the above database with those containing the hospital discharge records and the outpatient drug prescription records is a further strength. The latter databases contain data from every single hospital discharge carried out in the given period and of any pharmaceutical prescription issued by primary care physicians in the whole catchment area and in the examined period. This allowed us to collect data on exposures from earlier years without the risk of recall bias that could lead to differential misclassification.

Nonetheless, some limitations have to be considered. 

First, we had to restrict the analyses to only a few relevant comorbidities because data derived from the hospital discharge record database were incomplete. Therefore, we could not exclude residual confounding. 

Second, our sample included only adult patients, which could limit our findings’ generalizability to younger adults or children.

Third, in this study, we had information on the exact date of the diagnosis of resistance for both cases and controls, but it is possible that the outcome (namely, the onset of the resistance) preceded the exposure in some of the cases (reverse causality). For this reason, as in all case–control studies, interpreting the associations we found as causal should be carried out with caution. At the same time, it is likely that the risk of reverse causality is low in our case, as multiple bacterial cultures were carried out for many of the included patients, and the less recent was used to define the cases and controls. 

Fourth, the DDD gives a rough estimate of drug consumption, dose and duration of the treatment. 

Fifth, we assumed that the antibiotics prescribed were actually taken. This could have led to an overestimation of the exposure. 

Finally, we had no information on privately purchased antibiotics in our sample, which could have led to a non-differential underestimation of the exposure. However, according to aggregated data published annually by the Italian Medicine Agency, in Italy, privately purchased FQs that cannot be tracked account for less than 2% of total outpatient antibiotic consumption [29].

### 3.4. Meaning of the Study and Implication for Practice and Policy

*E. coli* isolates from individual patients with previous primary care prescriptions of FQs are more likely to be resistant to FQs than those collected from patients without. A single course of FQ is sufficient to increase the risk of resistance for up to two years. The risk for a patient to carry FQ-resistant *E. coli* strains is higher with more courses of previously prescribed FQs. Primary care clinicians may consider these findings, consistent to current recommendations, as a further reason to avoid unnecessary FQ use whenever possible. 

### 3.5. Implications for Future Research

The factors contributing to bacterial resistance to antimicrobials are diverse, with antibiotic prescribing in primary care being only one of its causes. Future studies should be designed to evaluate the individual risk of resistance of different bacterial species to different antibiotics, controlling, in a one-health perspective, also for additional factors other than primary care prescribing, such as prescribing in hospitals, wastewater treatment and intensive farming. Prospective designs, especially randomized controlled trials, rather than ecological studies, should be adopted to assess the impact of primary care antibiotic stewardship interventions on resistance rates of different bacterial species in individual patients over time to better assess causality. 

## 4. Methods

### 4.1. Study Design, Setting and Data Sources

This case–control study was conducted in the Autonomous Province of Bolzano, which, on the date of 1 January 2016, accounted for 525,475 inhabitants [30]. As data sources, the following information systems were anonymously linked to each other:The database of hospital reference laboratories was used to define cases and controls.The database of outpatient pharmaceutical prescriptions was used to define the exposure.The hospital discharge record database was used to identify potential risk factors.

From the database of hospital laboratories, we extracted all patients admitted to one of the local hospitals in 2016 and for whom a bacterial culture test was carried out. Analyses were conducted on blood, urine, respiratory tract secretions, soft tissue specimens and other samples (including vulvar, vaginal and perianal specimens; ascites and other abdominal fluid; pleural liquid; and post-surgery drainage fluid). The VITEK II system (bioMérieux, Hazelwood, MO, USA) or Maldi-TOF was used to identify bacterial species, and VITEK II was also used to perform antibiotic susceptibility testing. Following the EUCAST expert rules, ciprofloxacin was used as indicator antibiotic to detect FQ resistance in *E. coli* [31]. The yearly updated EUCAST interpretation criteria (Available online: http://www.eucast.org/ (accessed on 12 May 2022)) were used to interpret antibiograms, and specimens were either classified as resistant (R), susceptible (S) or intermediate (I). Only patients carrying FQ R or S *E. coli* isolates were included in the study. 

From the database of outpatient pharmaceutical prescriptions, we extracted all prescriptions issued from 1 January 2012 to 31 December 2016 by primary care physicians located in the Province of Bolzano (general practitioners and out of hours primary care physicians). We categorized antibiotics through the Anatomical Therapeutic Chemical (ATC) (Available online: https://www.whocc.no/ (accessed on 29 April 2022)) classification system (Appendix A) and used the cumulative defined daily doses (DDDs) of different drug classes prescribed in the five years preceding the “index date” (ID) as a comorbidity measure (Appendix A) [32,33].

From the hospital discharge records database, consistently with a recent literature review [23], we extracted the following potential confounding factors: age; gender; total days of hospitalizations; hospitalization with surgery, with device implantation or with organ transplant; and diagnosis of chronic diseases (cancer, diabetes, chronic obstructive pulmonary disease (COPD) and hemodialysis). 

Potential confounding factors and their data sources are listed in Table 4.

### 4.2. Definition of Cases and Controls

Patients were classified as cases if they were diagnosed with FQ-resistant isolates or as controls if they were diagnosed with FQ-susceptible isolates. We matched all available controls to each case, using the ID as the matching variable, defined as the day (±30 days) on which the culture test results became available. If the ID was not available, data were excluded from the analysis. If, during 2016, a patient had more than one culture test result, the less recent one was included in the analyses. The flow of included cases and controls is outlined in Figure 1.

### 4.3. Definition of Exposure

We defined the exposure of interest as the use of any FQ for systemic use (ATC codes are listed in Appendix A), grouped in the following three categories: (1) use of one or more FQ for systemic use in the first year preceding the ID, (2) use of one or more FQ for systemic use in the second year preceding the ID, and (3) use of one or more FQ for systemic use three to five years preceding the ID. In order to calculate the dose–response effect, exposed subjects were considered only those to whom at least one FQ was prescribed in the year preceding the ID, taking the number of prescriptions into account (0, 1, 2, 3 or more). 

### 4.4. Statistical Analysis

We used descriptive statistics to compare the characteristics of included patients, presenting categorical variables as percentages and continuous variables as mean (± standard deviation) or, where appropriate, as median (interquartile range). We used univariate conditional logistic regression to assess the strength of the association between single independent variables and the outcome. We used matched odds ratios (ORs) and 95% confidence intervals (CIs) to evaluate the strength of any association. We considered those factors with *p*-values < 0.05 in the univariate analysis as eligible for inclusion in the multivariable models. Covariates to be included in the regression models were then selected using a backward stepwise approach. 

### 4.5. Sensitivity Analyses

To evaluate the consistency of our results, we carried out the following sensitivity analyses: (a) in order to exclude all hospital-acquired infections from the outcome measure, we excluded from the analyses all patients with a culture carried out more than 48 h after their hospital admission; (b) in order to eliminate the effect of a very recent antibiotic use, we excluded all patients who received at least one prescription of antibiotics in the 15 days preceding the ID. 

All the analyses were performed using STATA software package version 17.0 and R 3.6 [34,35].

## Figures and Tables

**Figure 1 antibiotics-11-00822-f001:**
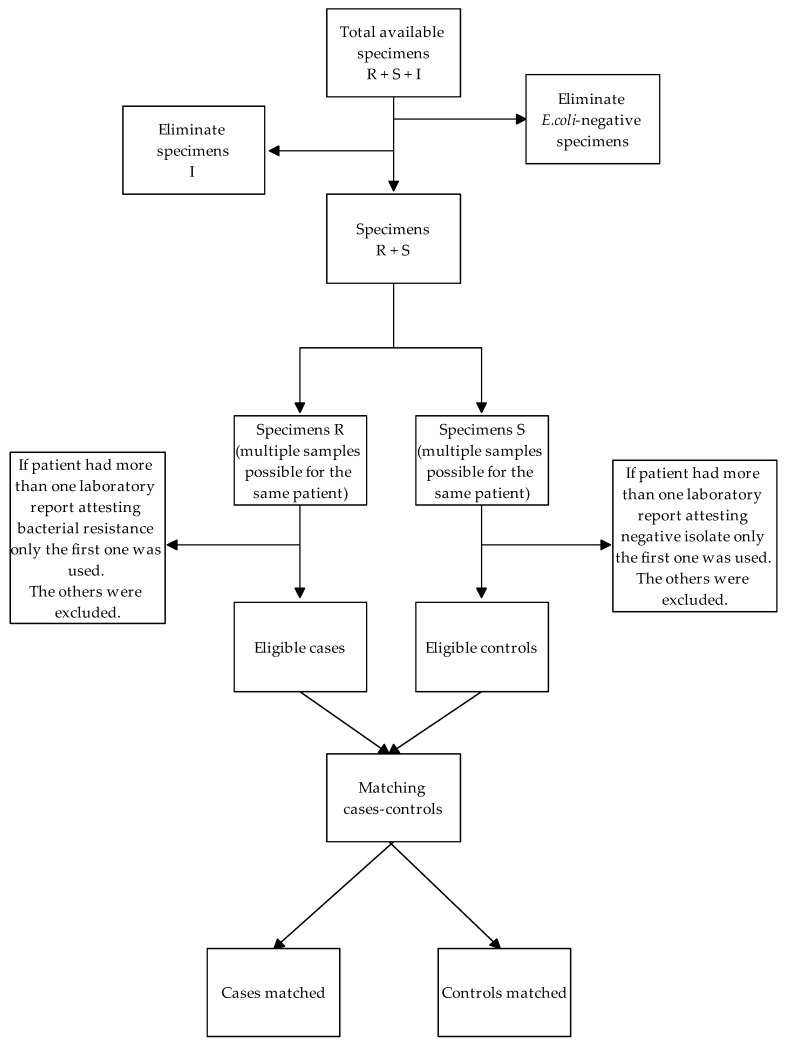
Flow of included cases and controls.

**Table 1 antibiotics-11-00822-t001:** Characteristics of case patients colonized/infected with QREC and matched control patients colonized/infected with FQ-susceptible *E. coli* and univariate conditional logistic regression analyses.

Variable	Cases ^a^ N = 403	Controls ^a^ N = 933	Crude OR (95% CI)	*p*
Age, Median (IQ)	78 (68–85)	74 (59–84)	1.17 ^b^ (1.10–1.25)	<0.0001
Gender, Male (%)	176 (43.03)	291 (31.19)	1.66 (1.30–2.11)	<0.0001
Drug’s DDD taken in previous 5 years, Median (IQ)	4760 (1741–8074)	2869 (256–6190)	1.07 ^c^ (1.04–1.10)	<0.0001
Number of active ingredients taken in previous 5 years, Median (IQ)	17 (9–24)	10 (4–18)	1.05 (1.04–1.06)	<0.0001
At least one FQ in previous 1st year (%)	161 (39.36)	148 (15.86)	3.87 (2.88–5.18)	<0.0001
At least one FQ taken in previous 2nd year (%)	129 (31.54)	142 (15.22)	2.72 (2.04–3.64)	<0.0001
At least one FQ taken in previous 3rd, 4th or 5th year (%)	173 (42.30)	261 (27.97)	1.90 (1.49–2.44)	<0.0001
FQ prescriptions in previous year (%) 0	248 (60.64)	785 (84.14)	Ref.	
1	72 (17.60)	83 (8.90)	3.14 (2.17–4.53)	<0.0001
2	40 (9.78)	35 (3.75)	3.80 (2.33–6.19)	<0.0001
3+	49 (11.98)	30 (3.22)	6.00 (3.55–10.17)	<0.0001
Number of hospitalizations in previous 5 years, Median (IQ)	4 (2–8)	2 (0–4)	3.67 (2.76–4.88)	<0.0001
Hospitalization days, Median (IQ)	48 (12–116)	10 (0–41)	1.07 ^d^ (1.05–1.09)	<0.0001
Hospitalization with surgery (%)	206 (50.37)	370 (39.66)	1.54 (1.23–1.95)	<0.0001
Hospitalization with device implantation (%)	44 (10.76)	65 (6.97)	1.57 (1.05–2.36)	0.029
Hospitalization with organ transplant (%)	9 (2.20)	18 (1.93)	1.19 (0.53–2.66)	0.673
Diagnosis of chronic diseases (%) Cancer	92 (22.49)	156 (16.72)	1.48 (1.11–1.98)	0.008
Diabetes	108 (26.41)	146 (15.65)	1.90 (1.44–2.51)	<0.0001
COPD	166 (40.59)	244 (26.15)	2.01 (1.56–2.59)	<0.0001
End-stage renal disease	10 (2.44)	15 (1.61)	1.67 (0.75–3.72)	0.213

^a^ Number (%) of patients or median (IQ); ^b^ OR calculated for 10-year increments; ^c^ OR calculated for 1000-DDD increments; ^d^ OR calculated for 10-day increments. Abbreviations: QREC = quinolone-resistant *E. coli.*

**Table 2 antibiotics-11-00822-t002:** Multivariable conditional logistic regression analysis focused on the association between previous FQ use and FQ resistance in *E. coli* over a five-year period.

Variable	Adjusted OR (95% CI)	*p*
At least one FQ prescription in 1st previous year	2.67 (1.92–3.70)	<0.0001
At least one FQ prescription taken in previous 2nd year	1.54 (1.09–2.17)	0.015
At least one FQ prescription taken in previous 3rd, 4th or 5th year	1.09 (0.80–1.48)	0.997
Age	1.09 ^a^ (1.01–1.18)	0.026
Gender, male	1.42 (1.07–1.88)	0.016
Hospitalization days	1.03 ^b^ (1.01–1.06)	0.022
Diagnosis of chronic diseasesDiabetes	1.41 (0.96–1.80)	0.037
COPD	1.43 (1.05–1.87)	0.019

^a^ OR calculated for 10-year increments; ^b^ OR calculated for 10-day increments. Abbreviations: FQ = fluoroquinolone, COPD = chronic obstructive pulmonary disease.

**Table 3 antibiotics-11-00822-t003:** Multivariable conditional logistic regression analysis focused on the association between FQ use and FQ resistance in *E. coli* over a 12–month period (dose-response effect).

Variables	Adjusted OR (95%CI)	*p*
FQ prescription in previous year 0	Ref.	Ref.
1	2.40 (1.62–3.56)	<0.0001
2	2.76 (1.63–4.66)	<0.0001
3+	4.21 (2.38–7.50)	<0.0001
At least one other J01 prescription in previous year	1.10 (0.83–1.45)	0.516
Age	1.11 ^a^ (1.03–1.20)	0.008
Gender	1.39 (1.05–1.84)	0.010
Hospitalization days	1.03 ^b^ (1.01–1.06)	0.020
Diagnosis of chronic diseases Diabetes	1.40 (1.02–1.93)	0.037
COPD	1.46 (1.09–1.96)	0.004

^a^ OR calculated for 10-year increments; ^b^ OR calculated for 10-day increments. Abbreviations: FQ = fluoroquinolone, COPD = Chronic obstructive pulmonary disease.

**Table 4 antibiotics-11-00822-t004:** Potential risk factors for FQ resistance in *E. coli* and their data source.

Potential Confounding Factor	Data Source
Age	hospital discharge records database
Gender	hospital discharge records database
Drug’s DDD taken in previous 5 years	database of drug prescription records
Number of active ingredients taken in previous 5 years	database of drug prescription records
Number of antibiotics taken in previous 5 years	database of drug prescription records
One or more J01 prescription taken in previous 5, 4 and 3 years	database of drug prescription records
One or more J01 prescription taken in previous 2 years	database of drug prescription records
Hospitalization days	hospital discharge records database
Hospitalizations	hospital discharge records database
Hospitalizations with surgery	hospital discharge records database
Hospitalizations with device implantation	hospital discharge records database
Hospitalizations with organ transplant	hospital discharge records database
Diagnosis of chronic diseases	hospital discharge records database
Cancer	
Diabetes Mellitus	
COPD	
Hemodialysis	

Abbreviations: DDD = defined daily dose; COPD = chronic obstructive pulmonary disorder.

## Data Availability

Raw data are available upon reasonable request from the corresponding author.

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
