# Peer review of "Effect of Fluoroquinolone Use in Primary Care on the Development and Gradual Decay of Escherichia coli Resistance to Fluoroquinolones: A Matched Case-Control Study"

_antibiotics, 2022, doi:10.3390/antibiotics11060822_

Round 1

Reviewer 1 Report

This manuscript by Peter Konstantin Kurotschka describes interesting data on the determinants of fluoroquinolone resistance in E. coli strains isolated in primary care. 

This manuscript is of major interest because of the major impact of antibiotic resistance in primary care.

The manuscript should be revised based on these comments before possible acceptance for publication.

Global: Prefer passive voice / Italicize bacterial names, "e.g.".

Results: A flow chart of included patients would be interesting to understand the impact of the selection process.

Table: Do not present non-significant p-values. Indicate significantly associated parameters in bold type.

Methods: the version of the EUCAST guidelines should be indicated.

Methods: table S3 should be reported in the main manuscript.

Methods: discuss how the authors planned to limit multiple testing biases in their study?

Reviewer 2 Report

This particular topic of bacterial resistance to currently available first-line antibiotics is of great importance since this phenomenon is one of the major threats to both human health and the healthcare system economy. Thus, I would like to congratulate the authors for the piece of knowledge provided through their research study.

However, I have found some issues needed to be considered, as stated below:

  1. Could you please reformulate the abstract, to clarify the aim of the present research study?
  2. line 77 - I think the proper word is ''analysis''
  3. line 91 - as you have highlighted, the antibiotic resistance development is strongly influenced by the length of hospitalization; In this setting, have you performed a comparative analysis between different pathogenic strains of E.coli? Is there a difference between hospital-acquired and community-acquired urinary tract infections? Do the involved bacterial strains have the same susceptibility to FQs?
  4. line 94 - could you please mention the method employed for the diagnosis of bacterial resistance to fluoroquinolones? If antibiograms were performed, could you please mention if there was a difference between molecules belonging to the fluoroquinolone class?
  5. line 122 - You have mentioned that older age represents an additional risk factor for antibiotic resistance development; As illustrated in Table 1, the age of patients included in the present study varies between 68 and 85 years old. Thus, the results of your research study apply only to the geriatric population? 
  6. line 142 - please correct ''whit''

Author Response

We thank the reviewer for his constructive feedback.  Point-by-point responses are listed below.  

Could you please reformulate the abstract, to clarify the aim of the present research study?

We changed the abstract accordingly (see line 21 and followings)

line 77 - I think the proper word is ''analysis''

Agreed and changed accordingly.

line 91 - as you have highlighted, the antibiotic resistance development is strongly influenced by the length of hospitalization; In this setting, have you performed a comparative analysis between different pathogenic strains of E.coli?

This would have been indeed interesting, but our aim was not to compare different pathogenic strains to compare different risk factors for different resistance mechanisms in E. coli, rather we aimed to uncover the effect of previous FQ use on the development and decay of FQ-resistance. We used routinely collected biosamples, in which only routine microbiological analyses were performed following current best laboratory practice.  

Is there a difference between hospital-acquired and community-acquired urinary tract infections?

We were interested mainly in community acquired infections (mainly, but not only, of the urinary tract). By definition, hospital acquired infections are those diagnosed through a microbiological test performed >48 hours from hospital admission. Therefore, we performed a sensitivity analysis to exclude all patients in whom a bacterial culture was performed >48 hours from hospital admission, which results were consistent with the main analysis (supplementary table S4, mentioned in line 111-114 and 291-297)

Do the involved bacterial strains have the same susceptibility to FQs?

Please cf. answer to your comment regarding line 91 and the following answer to your comment (line 94).

line 94 - could you please mention the method employed for the diagnosis of bacterial resistance to fluoroquinolones? If antibiograms were performed, could you please mention if there was a difference between molecules belonging to the fluoroquinolone class?

We used routinely collected laboratory data. Based on the EUCAST guidelines, laboratories perform FQ susceptibility tests using ciprofloxacin. If the bacterial strain is resistant to ciprofloxacin, the EUCAST expert rule states that the strain is resistant to all fluoroquinolones, if not, it is deemed to be sensible. We added this information (lines 244-248) and cite the EUCAST expert rule (ref. 31).

line 122 - You have mentioned that older age represents an additional risk factor for antibiotic resistance development; As illustrated in Table 1, the age of patients included in the present study varies between 68 and 85 years old. Thus, the results of your research study apply only to the geriatric population? 

We thank the reviewer for this relevant comment. We now address the issue in the limitation section (line 191-2).

Round 2

Reviewer 1 Report

The manuscript has been revised according to my previous comments.

Reviewer 2 Report

Thank you for considering my suggestions.

As all the issues outlined were properly addresed, I think the present manuscript is worth being published in Antibiotics.